# Self-Supervised Pretraining for Large-Scale Point Clouds

**Zaiwei Zhang**
AWS AI
Santa Clara, CA 95054
zaiweiz@amazon.com

**Min Bai**
AWS AI
Santa Clara, CA 95054
baimin@amazon.com

**Erran Li**
AWS AI
Santa Clara, CA 95054
lilimam@amazon.com

## Abstract

Pretraining on large unlabeled datasets has been proven to improve the downstream task performance on many computer vision tasks, such as 2D object detection and video classification. However, for large-scale 3D scenes, such as outdoor LiDAR point clouds, pretraining is not widely used. Due to the special data characteristics of large 3D point clouds, approaches for 2D pretraining frameworks tend to not generalize well to this domain. In this paper, we propose a new self-supervised pretraining method that targets large-scale 3D scenes. We pretrain commonly used point-based and voxel-based model architectures and show the transfer learning performance on 3D object detection and semantic segmentation. We demonstrate the effectiveness of our approach on both dense 3D indoor point clouds and sparse outdoor LiDAR point clouds.

## 1   Introduction

Accurate and reliable perception is core to the capabilities and safety of autonomous and robotic systems such as self-driving vehicles and indoor robotics. To accurately measure the structure of their surroundings, such platforms usually rely on LiDARs and depth cameras for outdoor and indoor scenes, respectively. Given the 3D point clouds, models are built for a variety of perception tasks, such as object detection, scene segmentation, localization, and mapping.

Modern deep learning models have shown great capabilities when large quantities of annotated data are given. However, generating the required amount of annotations is a laborious and expensive manual process [14; 57; 18]. This is especially true for labeled datasets targeting object detection and semantic segmentation of large real-world scenes, which can contain dozens of independently moving objects and irregularly shaped background material. Moreover, the 3D data is difficult to clearly visualize, manipulate, and annotate on a 2D screen. In the case of outdoor LiDAR scans, the annotation tasks are further complicated by the sparsity of the typical point cloud. However, the cost of acquiring a large quantity of unlabeled sensor data is usually much lower compared to manual annotation. Hence, developing algorithms to automatically discover useful patterns and strengthen the performance of models with few annotated examples is of great value.

Recently, researchers have explored self-supervised learning (SSL) to bootstrap model training in various domains. In computer vision, [26; 8; 22; 25] make use of datasets such as ImageNet [50] with carefully designed pretext tasks to learn useful features for downstream tasks such as object detection and segmentation. These techniques rely on the enormous ImageNet dataset, which depicts

36th Conference on Neural Information Processing Systems (NeurIPS 2022).

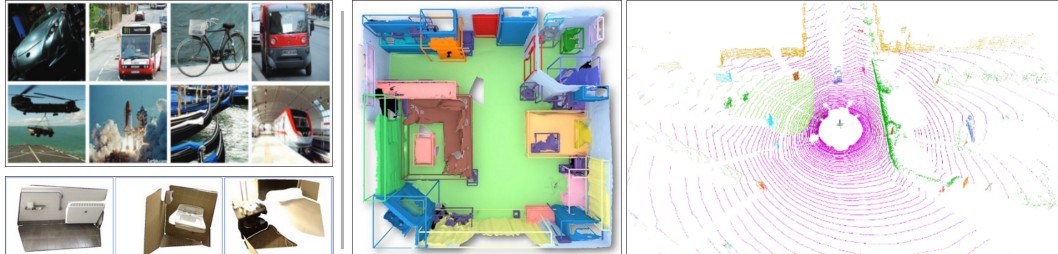

**Figure 1:** Complexities of datasets. Other techniques pretrain on simpler scenes like ImageNet [50] (upper left) where each image prominently features one foreground object or single ScanNet [14] frames (lower left). On the other hand, we target highly complex indoor scenes like aggregated complete ScanNet scenes (middle) and LiDAR scans of road scenes like SemanticKITTI [5] (right) with dozens of different objects (colored).

tens of thousands of object types. Most images prominently feature a single foreground object, which can be well-described by a global feature vector for each sample. In the 3D point cloud domain, several works [72; 65] have proposed pretraining techniques to learn useful features for downstream tasks. However, these methods likewise primarily focus on much simpler scenes, such as single objects or a limited indoor scene captured by a depth camera from a single viewpoint.

On the other hand, we target complex 3D scenes with a multitude of objects and background material. We visualize the differences between the subject datasets of existing approaches and our method in Figure 1. We first consider large-scale, complete 3D indoor scene models generated from combined depth images. Next, we explore outdoor environments with LiDAR points up to 100m away on dozens of objects and complex background regions. We refer to our method as SSPL (**s**elf **s**upervised **p**retraining for **l**arge-scale point clouds). Unlike most of the prior work, we propose a technique which performs contrastive learning on local features extracted from sub-regions (volumes) within large-scale point clouds. Our algorithm makes use of a local contrastive loss within a scene and as well as a global contrastive loss across different scenes. As we will demonstrate, our method involving local feature reasoning greatly outperforms the existing methods in our settings.

In practice, [71] has noted that there are significant domain gaps attributable to the LiDAR sensor due to its sparsity, even when the observed scenes and downstream tasks are highly similar. Motivated by this, we explore the setting of in-domain self-supervised pre-training for LiDAR point clouds. As we will demonstrate, existing 3D SSL methods [72; 65] are unsuitable for highly complex outdoor scenes, motivating the need for new algorithms.

In this paper, we make the following contributions:

- We propose a self-supervised pretraining method which leverages reasoning over local volume level features for 3D point clouds.
- We introduce a novel ClusterInfoNCE loss to efficiently perform contrastive learning globally for a large set of local volume features.
- We show that our pretraining approach outperforms baselines on multiple downstream tasks on both single-view outdoor LiDAR and large-scale indoor point clouds, with up to +6.2% higher AP25 in object detection and +3.6% mIoU in semantic segmentation.

## 2    Related work

Our method builds on and extends the work from the self-supervised learning literature, with a focus on large-scale 3D point clouds. In this section, we give an overview of the recent advances in self-supervised learning, as well as perception and representation learning in 3D.

**Self-supervised learning for images** Self-supervised learning has been studied extensively in machine learning and computer vision [61; 44; 49; 51; 39]. There are many approaches for 2D representation learning, such as clustering [6; 7; 32], GANs [16; 40], various pretext tasks [15; 42; 62] *etc*. Recent advances [41; 26; 28; 8; 10; 59; 35; 20; 22; 25] have shown that self-supervised pretraining exhibits great advantages in transfer learning, especially for benchmark with limited labels. [9; 4] both use similar local and global contrastive learning approaches but [9] uses domain specific

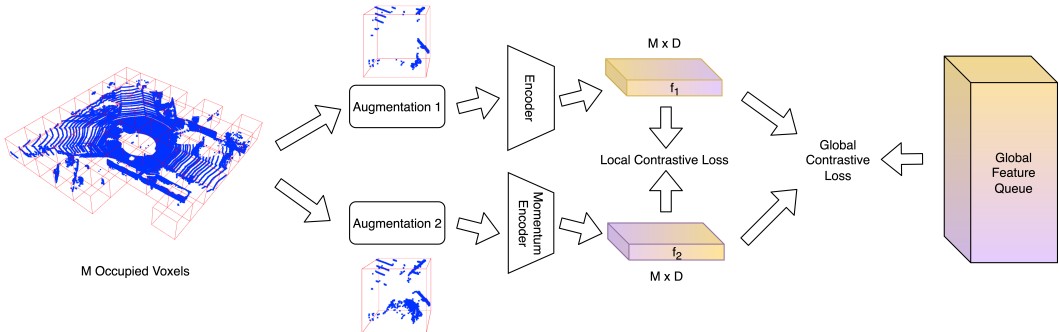

**Figure 2:** Overview of SSPL- a 3D representation learning method for large-scale point clouds pretraining. We split a 3D scene point cloud into $M$ occupied volumes and apply two different data augmentations for point clouds in each volume. In this figure, we show data augmentation examples of a single volume. We use a neural network encoder and a momentum encoder to extract spatial features which are pooled and projected to obtain local volume-level features. With paired volume features, SSPL optimizes the local contrastive loss between paired features and optimizes a global contrastive loss between the local features and the features in the global feature queue.

clustering for their global feature learning and [4] mainly studies the local feature affinities while we explicitly conduct similarity measures across different instances. [27] conducts data augmentation on local features but not global reasoning across different images. Our work adopts the momentum encoder from *MoCo* [26] to extract features for instance discrimination, instead of a memory bank.

**Self-supervised learning for 3D data** Studies on unsupervised feature learning for 3D data have mainly been focused on single 3D object representation with applications on reconstruction, classification or part segmentation [2; 17; 23; 24; 37; 52; 63; 69; 34; 1]. However, due to the domain gap issues, features learned from single 3D object representations does not transfer well to scene level understanding tasks [65]. Recently, there has been a growing interest in designing self-supervised methods to build representations of scene level point clouds [65; 31; 72]. Xie *et al*. [65] proposed a self-supervised method based on enforcing contrastive learning on point correspondences between different views of 3D scenes. These point-wise correspondences can be acquired after aligning different depth maps into a unified 3D scene, which requires time-consuming 3D registration algorithms or specific human labeling efforts on identifying static correspondences. In the case of sparse outdoor LiDAR point clouds, temporal correspondences are often not well defined. Similarly, another line of work [31; 19] rely on multiple scene scans' alignments. On the other hand, our method only requires single frame 3D scans, making it more generally applicable. Zhang *et al*. [72] conducted an instance discrimination task on global scene-level features while our method focuses on reasoning across local patches.

**Supervised 3D perception** With the significant applications of robotics and self-driving vehicles, 3D perception has been a widely studied topic. Among the many perception tasks, supervised object detection [46; 54; 67; 53; 55; 36; 70; 11; 68; 73] and semantic segmentation [74; 60; 66; 29; 30; 33] have been explored by a large number of prior works. Because of the popularity and usefulness of these tasks, we select them as the downstream tasks to analyze the effectiveness of our approach. However, we note that as our self-supervision is task agnostic, it can be applied to other downstream tasks. To achieve these tasks, a number of neural network backbones have been proposed to process 3D point cloud data, which is usually given as an unordered and irregular set of points [47; 21; 58]. In our work, we use both the popular PointNet++ [48] based on pointwise reasoning and 3D U-Net [66] based on sparse voxel convolutions to demonstrate the general effectiveness of our approach. Recently, [43] has proposed a similar local contrastive learning approach on clustered point clouds but they did not conduct global reasoning for those local point clusters.

## 3 Method

We introduce our self-supervised pretraining framework: SSPL as outlined in Algorithm 1, and visualized in Figure 2. As shown in Figure 1, we focus on large-scale point clouds, which contain numerous objects and distinct structures within each scene. In these cases, a global vector encoding requires a combinatorial and intractable level of expressiveness to describe the attributes and

---

**Algorithm 1** Training Framework for SSPL

---

**Input** Backbone architecture NN; Dataset $\mathcal{D} = \{\mathbf{X}\}_{i=1}^N$; Global feature queues $F_g \in \mathrm{R}^{C \times D}$;
**Output** Pre-trained weight for the NN
   **for** $x_i$ in $X$ **do**
       - Split $x_i$ into $M$ occupied volumes: $x_i = \{V\}_{i=1}^M$
       - From $x_i$, generate two augmented versions $x_i^1$ and $x_i^2$
       - Generate *local* crops $v_i^1$ in $x_i^1$ and *global* crops $v_i^2$ in $x_i^2$
       - Compute volume features $f^1, f^2 \in \mathrm{R}^{M \times D}$
       - Compute local contrastive loss $L_c(f^1, f^2)$ and global contrastive loss $L_g(f^1, f^2, F_g)$
       - Update $F_g$ with $f^2$ and update NN
   **end for**

---

locations of the various objects. Therefore, we propose to perform representation learning on local features. We first describe local volume features extraction in Section 3.1, and introduce the local contrastive loss in Section 3.2. Finally, we discuss the formulation of the global contrastive reasoning in Section 3.3.

### 3.1 Local feature extraction

Given a large scale 3D point cloud instance $x_i$, we split it into $M$ occupied volumes. Similar to patches in 2D images, we can predefine a volume size and sample up to $M$ occupied volumes based on typical scale of input scenes. $x_i$ is then transformed into two augmented views $x_i^1$ and $x_i^2$ by applying random rotation and scaling. We then sample points in each occupied volume and generate two crops with different sizes for two augmented views. The smaller crops $v_i^1$, generated from $x_i^1$, are defined as *local* crops while larger ones $v_i^2$, generated from $x_i^2$, are *global* crops. This scheme creates natural local-global correspondences, which enables the network to predict global features from local information in feature learning. $x_i^1$ and $x_i^2$ are mapped to feature vectors by applying a neural network. After acquiring the per-point/per-voxel features, we group the features by their corresponding volumes and apply max-pooling to the last layer's features. We then apply a two layer MLP as in [10] and L2 normalization to compute the volume features $f^1$ and $f^2$.

**Momentum encoder** Adapting the method of He *et al.* [26], we use an encoder $\theta_s$ to extract features for $x_i^1$ and a momentum encoder $\theta_t$ to extract features for the augmented view $x_i^2$. The weights of the momentum encoder are updated with an exponential moving average (EMA) from the encoder's weights. The update rule is $\theta_t \leftarrow \lambda\theta_t + (1 - \lambda)\theta_s$ with a fixed $\lambda = 0.999$.

### 3.2 Local volume reasoning

Traditionally, given a pair of volume features $f^1, f^2 \in \mathrm{R}^{M \times D}$ from two augmented versions of $x_i$ instance, feature consistency is usually applied as a pre-text task. For contrastive learning approaches, InfoNCE [26] is usually enforced, where volume features $f_1$ and $f_2$ are pushed to be similar and volume features $f_1$ are dissimilar to all other volume features from other instances. However, computation costs increase linearly as the number of feature embedding increases for each batch. Traditionally, InfoNCE assumes that there is one feature embedding for each data instance. Since in average there are about 300 occupied volumes for each scene in our experiments, it is computationally infeasible to perform InfoNCE on each volume feature with a large set of global features as negative examples.

Therefore, we adapt the PointInfoNCE [65] loss to conduct contrastive learning between different volumes of the same point cloud. Compared to PointInfoNCE, we reason about volume-level information, where the loss is defined as:

$$L_c = - \sum_{j \in M} \log \frac{\exp(f_j^1 \cdot f_j^2 / \tau)}{\sum_{k \in M} \exp(f_j^1 \cdot f_k^2 / \tau)} \tag{1}$$

Here, $M$ is the set of all occupied volumes for the $x_i$ instance, and $\tau$ is a temperature parameter [64]. The positive examples are the two augmented crops from the same volume, while negative examples

are from other volumes under the same instance. In practice, we find that performing this local volume reasoning alone as a pre-text task gives comparable or better pretrained models compared to baseline methods. Please refer to Section 5.4 for experimental evidences.

Although applying BYOL [22] as a pretext task does not yield computational issues, we find that it results in minimal performance gains in our settings. Please see Section 5.3 for more details.

## 3.3 Global volume reasoning

One natural objective for global contrastive reasoning is to use instance discrimination on volume features. As mentioned before, this objective does not scale well with large numbers of occupied volumes. Across different large scale scene datasets, we observe that there are common objects appearing in almost every scene, *e.g.* road and car in outdoor scenes or chair and sofa in indoor scenes. Inspired by this, we propose to conduct global volume reasoning through a fixed number of global clusters. This reduces the computational burden and adds regularization during pretraining.

**Preliminary on unsupervised feature clustering** Traditional methods on feature clustering, such as k-means and DBSCAN have proven to be effective but time-consuming to apply on large dataset. Recently, a line of methods [3; 8] show that pseudo-label assignment can be formulated as an optimal transport problem, where clustering can be done efficiently by the Sinkhorn-Knopp algorithm [13]. We follow a similar approach for our feature clustering steps.

**Global feature queue** We first build a global feature queue $F_g$ to store features of occupied volumes over multiple data instances. Adapting from He *et al.* [26], the global feature queue is created with volume features $f_2$ from the momentum encoder. The following steps are then used during training.

**Global feature clustering** For building global clusters, we follow the same method proposed by Asano *et al.* [3]. Given global feature queue $F_g \in \mathrm{R}^{C \times D}$, where $C$ is the queue size and $D$ is the feature dimension, we apply a classification head $h_g : \mathrm{R}^D \to \mathrm{R}^K$ to convert the global feature vectors to class (cluster ID) scores. We then map them to class probabilities via a softmax operator: $P = softmax(F_g \cdot h_g)$. We find the class label assignments $Q$ by performing iterative Sinkhorn-Knopp updates [13].

**Global contrastive loss** Given a pair of volume features $f^1, f^2$, we query the global feature queue with local features $f^2$ to extract the corresponding cluster labels for each volume. We use kNN on $l2$ feature distances when querying, and we set $k = 1$ for clustering.

With volume level class labels, it is natural to use a classification pretext task for pretraining. However, compared with pretraining on object-level features (*e.g.* ImageNet), our volume-level features usually lack class diversity and feature diversity. For indoor scenes, there are repeating wall and floor volumes, while a majority of volumes are roads and buildings in outdoors scenes. We observe that such pretraining strategy provides minimal improvements. (Please see section 5.4 for details.) To directly apply feature representation learning across scenes, we propose a **ClusterInfoNCE** loss:

$$L_g = -\frac{1}{K} \sum_{i=1}^{K} \log \frac{\exp(f_i^1 \cdot knn(f_i^2, F_g)/\tau)}{\sum_{j!=i} \exp(f_i^1 \cdot F_g^j/\tau)} \tag{2}$$

where $K$ is the number of clusters and $f_i^1$ and $f_i^2$ are drawn from $f^1$ and $f^2$ if their corresponding class labels are $i$. $knn(f_i^2, F_g)$ extracts the most similar features of $f_i^2$ from $F_g$, which serve as positive examples. For negative examples, $F_g^j$ are sampled from $F_g$ if their corresponding class labels are $j$. In our experiments, we find that using 20k negative examples works well. Since the runtime of **ClusterInfoNCE** loss depends only on number of clusters used, it scales well with more occupied volumes. By optimizing this loss, we are explicitly enforcing feature consistency across scenes and encouraging the volume features to form clusters globally. As shown in Section 4, we observe that adding the **ClusterInfoNCE** loss results in improvements on various downstream tasks. As well, we compare it with different negative sampling strategies in Section 5.4.

## 3.4 Implementation details

In our experiments, we set the size of global feature queue to be 300K. We use a temperature value of 0.1 while computing the non-parametric softmax in Eq 1 and 3. The local and global contrastive

**Table 1:** Fine-tuning performance on ScanNet Dataset and SUN RGB-D Dataset (AP25)

| Self-Supervision Method | % of Scan Used for Fine-Tuning | | | | % of SUN Used for Fine-Tuning | | | |
|---|---|---|---|---|---|---|---|---|
| | 20% | 50% | 70% | 100% | 20% | 50% | 70% | 100% |
| None | 46.1 | 52.3 | 54.0 | 58.6 | 45.2 | 55.2 | 55.6 | 57.4 |
| BYOL [22] | 48.2 | 54.0 | 55.5 | 59.1 | 47.7 | 55.6 | 56.4 | 58.0 |
| PointContrast [65] | - | - | - | 59.2 | - | - | - | 57.5 |
| DepthContrast [72] | 50.1 | 56.0 | 57.0 | 60.1 | 45.5 | 56.5 | 57.5 | 59.1 |
| **Ours** | **53.0** | **58.5** | **60.6** | **63.0** | **47.5** | **57.8** | **58.5** | **60.1** |

loss are equally weighted. For training, we use a standard SGD optimizer with momentum 0.9, and we use a cosine learning rate scheduler [38] which decreases from 0.06 to 0.00006 and train the model for 500 epochs with a batch size of 96.

# 4 Experimental results

In this section, we thoroughly analyze the efficiency of our proposed method. First, we describe the tasks and datasets involved in our experiments. Then, we present quantitative results, demonstrating the increase in performance relative to our comparison baselines. Finally, we take a closer look at the contributions of each of our proposed components.

The key goal of self-supervised learning is to automatically learn feature extractors that are useful for actual downstream tasks. Moreover, it is beneficial if the learned features are useful for different tasks, as this improves the usability of the technique and reduces the need for task-specific redesign. Our approach is chiefly designed for large scale scenes that are not dominated by a single, well defined foreground object. We demonstrate the effectiveness of our self-supervised learning approach by showing that downstream models can achieve significantly higher performance using a small fraction of available annotations. Next, we dive into the features learned by our model to gain additional insights, followed by various ablation studies to justify the design choices in our approach. As mentioned in Section 1, for LiDAR point clouds, the sensor configuration has a tremendous impact on the structure of perceived 3D points. Therefore, we opt to pretrain and finetune our models in each domain individually for outdoor datasets.

## 4.1 Baseline methods

We limit our input data to measurements taken at single instants in time (instead of temporal sequences) for generalization. As such, we use the recent BYOL [22] and DepthContrast [72] (based on MoCo [26]) as baselines. In both cases, we use the same full scene augmentation scheme as proposed in DepthContrast. For indoor scene pretraining, to introduce more scene complexity, we use the point clouds of fully reconstructed scenes built by aggregating multiple sequential measurements of the scene. Therefore, we also use PointContrast [65] as a comparison for the indoor setting.

## 4.2 Large scale dense point clouds

First, we evaluate our feature learning framework on point clouds produced with common depth cameras. In our work, we choose the ScanNet [14] dataset for pretraining. ScanNet contains 1500 large scale indoor scenes, with each scene consisting of a variety of objects. Additionally, it contains more than 10 different room types with diverse room scales. We believe that ScanNet covers a diverse set of indoor object distributions.

Compared to DepthContrast [72] and PointContrast [65], we only utilize the full scans from ScanNet training dataset for pretraining. We aim to learn sharp local features from large scale scenes, and thus we are only interested in feature learning with full views instead of partial views. The pretraining dataset contains 1201 full view scans with 50k points per scan. We use a PointNet++ [48] backbone network for pretraining, as it is widely used for different 3D perception benchmarks.

To evaluate the transfer learning performance of our pretrained models, we use two commonly used indoor 3D object detection benchmarks: ScanNet (SCAN) [14] and SUN RGB-D (SUN) [56]. We

finetune our pretrained model by using it as the backbone initialization of VoteNet [45] (a well known 3D object detector), and report the detection performance using the mean Average Precision at IoU=0.25 (AP25) metric. To study the label efficiency of our pretrained models, we also subsample different sets of the training data used for finetuning. We follow the setup in VoteNet [45] for finetuning.

Based on the results in Table 1, our approach significantly improves downstream object detection performance across all settings, with up to 4.4% improvement on ScanNet and 2.7% on SUN RGB-D. Moreover, on ScanNet, our pretraining approach together with 20% of labeled data allows the downstream detection task to achieve the equivalent performance of using 50% of labeled data and training from scratch. Practically, this can lead to significant savings in labeling cost. Our approach also outperforms the top-performing baseline method DepthContrast. (Note that we reproduce DepthContrast with the original Votenet model and pretrain only on the 1200 instances for fair comparison.) We see a more prominent performance gain on ScanNet compared to on SUN RGB-D, as the former contains only 1201 training scenes while the latter contains more than 5k.

## 4.3   Large scale sparse point clouds

Next, we further increase the size and complexity of the 3D scene by using outdoor scenes collected for autonomous vehicles using LiDAR units. These scenes consist of a large number of different objects and regions, spanning dozens of semantic classes. As we will show, it is critical to consider local regions separately in the self-supervised learning framework.

Our algorithm aims to learn useful features for all regions in a point cloud. While detectable individual objects cover a large fraction of the 3D structures present in the previous indoor datasets, compact and countable foreground objects suitable for detection such as cars and pedestrians consist of only 10% of a typical outdoor scene. As such, we select the semantic segmentation task to validate our method, as it aims to give a meaningful label to every 3D point captured by the sensor.

We use two large scale, well-established public datasets for this task, which contain a large variety of scenes. SemanticKITTI (SK) [5] is based on the original KITTI [18] dataset, which is one of the original autonomous driving datasets that propelled a tremendous amount of progress in the perception field. This dataset contains 19k training frames over 10 sequences with approximately 122k points per scene. The more recent Waymo Open Dataset (WOD) [57] provides a further leap in the scale and variability of data by providing 24k training frames sampled from 850 driving sequences, with a higher point density of 161k points per scene. Both datasets define approximately 20 semantic classes. For these experiments, we use the sparse 3D convolution backbone based on U-Net [12] as it is used in current state of the art semantic segmentation models [66; 74]. As well, we largely follow the training settings of [66].

To show that our approach extracts meaningful representations, we present semantic segmentation results on subsampled sets of the training data provided by SemanticKITTI and WOD. The samples are uniformly and randomly selected from the overall training datasets. To simulate the scenario where different levels of human annotation effort is available, the samples are drawn such that any smaller fractional subset is contained within any larger subset. The results are shown in Table 2. We add a light-weight decoder with two convolutional layers to the pretrained point-wise feature encoding backbone network, and finetune until convergence. In each case, we report the average performance over the checkpoints of the last 15% of the training process to reduce noise.

Our approach significantly improves downstream semantic segmentation when the annotation is limited. For example, we observe 3.6% and 2.3% improvements in mIoU on SemanticKITTI and WOD, respectively, when annotations for 1% subsets of the full datasets are provided. Moreover, these improvements are much more significant than those gained from pre-training with BYOL and DepthContrast, suggesting that beneficial pre-training on large scale point clouds require designs with specific considerations. On the other hand, we observe that the gains are reduced when additional annotated data is provided, as seen in the case with 5% and 10% annotations, as expected.

## 4.4   Feature analysis

Here, we examine the features learned by our self-supervision scheme. As with the other analysis in our work, we compare with the BYOL and DepthContrast baselines, and use the SemanticKITTI

**Table 2:** Fine-tuning performance on SemanticKITTI Dataset and Waymo Open Dataset (mIoU)

| Self-Supervision Method | % of SK Used for Fine-Tuning | | | | % of WOD Used for Fine-Tuning | | | |
|---|---|---|---|---|---|---|---|---|
| | 1% | 2% | 5% | 10% | 1% | 2% | 5% | 10% |
| None | 38.9 | 44.0 | **51.7** | 53.4 | 42.5 | 45.8 | 50.4 | 52.8 |
| BYOL [22] | 38.8 | 43.4 | 51.0 | 52.3 | 42.3 | 46.5 | 49.9 | 53.2 |
| DepthContrast [72] | 39.2 | 44.7 | 49.9 | 52.3 | 42.7 | 45.8 | 50.7 | 53.0 |
| Ours | **42.5** | **46.4** | 51.0 | **53.6** | **44.8** | **47.3** | **51.3** | 53.5 |

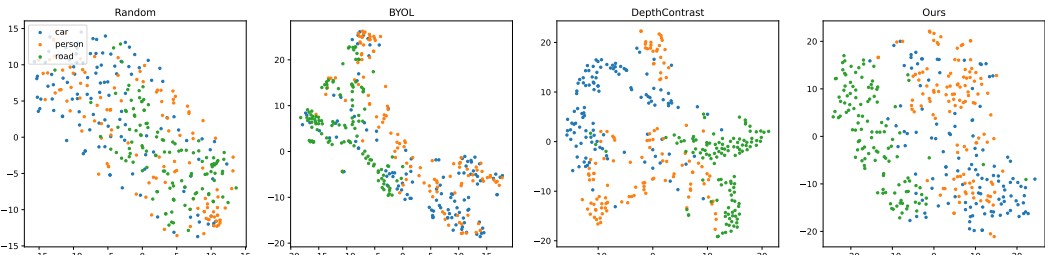

**Figure 3:** t-SNE visualization of features generated by backbone network with various weight settings.

dataset for the study. We visualize the final per-point features from the 3D UNet model using the t-SNE dimensionality reduction and color the points according to their semantic class. For clearer interpretability, we limit the semantic classes to the commonly seen and important classes of *road, person, and car*. The points are subsampled from the validation set.

We see that the embeddings produced using DepthContrast and our approach are both able to separate the three classes in feature space to a much stronger degree than the BYOL baseline, as well as the random initialization which serves as a sanity check. In comparison with DepthContrast, our embeddings have larger variance within the feature space while still being able to separate the semantic classes, suggesting that they can possibly preserve more distinctive geometric information while extracting meaning. This hypothesis is supported by the added performance gain of pretraining using our method compared with DepthContrast as seen in Table 2.

## 5 Ablation study

In this section, we demonstrate the importance of each component in our training framework. We conduct our ablation studies with models pretrained on the widely accepted SemanticKITTI dataset with complex and large-scale scenes in autonomous driving.

### 5.1 Volume cropping ratios

In the left side of Figure 4, we report the impact of applying different cropping ratios for the *local* crop. For the *global* crop, we randomly crop approximately 50% of the original volume so that most of the geometry feature is still preserved in the momentum encoder. We observe that smaller *local* crops produce marginal improvements in finetuning, and that a 10% crop ratio is optimal.

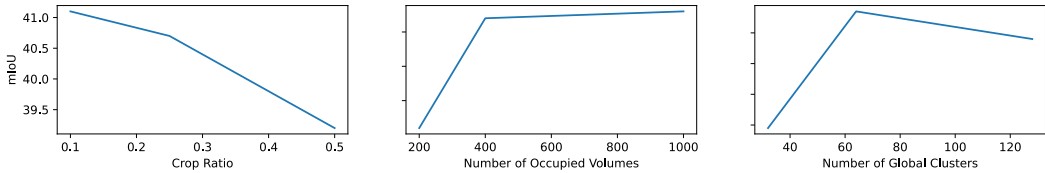

**Figure 4:** Segmentation ablation study on SemanticKITTI using 1% data for fine-tuning. Left: mIoU vs number of occupied volumes, middle: mIoU vs volume crop ratios, right: mIoU vs number of global clusters.

**Table 3:** BYOL vs contrastive methods

|        | Scratch | BYOL | Contrastive |
|--------|---------|------|-------------|
| 1% SK  | 38.9    | 39.8 | **41.7**    |

**Table 4:** Different global volume reasoning strategies

|        | Local | Cls  | Hard | Rand | Our  |
|--------|-------|------|------|------|------|
| 1% SK  | 41.7  | 40.8 | 41.0 | 40.5 | **42.9** |

## 5.2 Number of occupied volumes

For local volume reasoning, the number of occupied volumes plays an important role as it directly affects the number of negatives used for local contrastive loss. Intuitively, by increasing the number of occupied volumes, the local contrastive loss forces the network to learn more distinct local features. However, with a fixed model size, the expressiveness of local features saturates after it forms a certain number of clusters, which directly correlates with number of occupied volumes in our training approach. As shown in the middle of Figure 4, we see strong performance with 400 occupied volumes with little gain for further increases.

## 5.3 BYOL vs contrastive method

Instead of contrastive methods, Grill *et al*. [22] proposed a new approach for self-supervised learning **BYOL**, which directly enforces feature consistency between a student network and a teacher network. To better understand our framework, we reproduce the BYOL loss formulation to our best effort to replace the PointInfoNCE in the local volume reasoning. Formally, we use the following mean square error between the normalized predictions and target projections:

$$L_c^{'} = \sum_{j \in M} ||z_\theta(f_j^1) - f_j^2||^2 \tag{3}$$

Similar to Equation 2, $M$ is the set of all occupied volumes, and $z_\theta$ is a predictor for the student network. Based on the results from Table 3, the features learned with BYOL performs significantly worse than the contrastive approach in 1% SemanticKITTI finetuning. We hypothesize that within the same instance, some volumes share very similar feature patterns.

Without explicitly encouraging dispersion within the feature space (e.g. negative examples), BYOL learns less distinguishing features in our setting.

## 5.4 Different strategies for global volume reasoning

As mentioned in Section 3, a natural pretext task for global volume reasoning is classfication. Given a pair of volume features $f^1$, $f^2$ and their corresponding cluster labels $Q^v$, we apply a linear reprojection $h_l$ to map the volume features to class scores, and optimize the classification loss below:

$$L_{cls} = -\sum_{j \in M} Q_j^v \log P_j^1 - \sum_{j \in M} Q_j^v \log P_j^2 \tag{4}$$

where $P_j^1 = softmax(f^1 \cdot h_l)$ and $P_j^2 = softmax(f^2 \cdot h_l)$. As shown in Table 4 where **Cls** represents the model pretrained on the classification loss, we see that it does not improve the finetuning performance, compared to using local contrastive loss alone (**Local**). We also compare different negative sampling strategies for the ClusterInfoNCE loss. In our approach, after we find the positive feature example for cluster $i$, we sample negative feature examples from other clusters. In Table 4, we compare with only sampling negative examples from cluster $i$ (**Hard**) (which can be seen as hard negative sampling) as well as random sampling from all clusters (**Rand**). Based on the results, our approach (**Our**) provides the largest performance boost.

## 5.5 Number of global clusters

We explore the impact of the number of global clusters used in our global feature queue to compute the contrastive loss (right plot of Figure 4). This hyperparameter is shown to be impactful, where $64$ is the best setting for the SemanticKITTI LiDAR scans. We believe that the optimal number is influenced by the complexities of the 3D scenes in question.

# 6   Limitations, future work, and broader impact

Lastly, we discuss the limitations of our model. As the characteristics of 3D sensors vary widely (dense depth scans vs sparse LiDAR point clouds), our approach relies on in-domain unlabeled data as opposed to producing a general purpose foundation model like those used in 2D images and language. Moreover, we do not make use of any temporal information, which can provide significant cues for feature consistency. We believe that these are possible areas for future research. We believe that increasing 3D perception capabilities of autonomous agents will allow them to operate more safely and enrich our lives. As self-supervised learning is able to extract useful representation from unlabeled datasets, it can be less impacted by biases that may influence annotation processes, and possibly improve the reliability and fairness of perception models.

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
