# Self-Supervised Pretraining for Large-Scale Point Clouds Supplementary Material

**Zaiwei Zhang**
AWS AI
Santa Clara, CA 95054
zaiweiz@amazon.com

**Min Bai**
AWS AI
Santa Clara, CA 95054
baimin@amazon.com

**Erran Li**
AWS AI
Santa Clara, CA 95054
lilimam@amazon.com

## 1   Appendix

We first provide the model architecture details in Section 1.1. We then discuss the hyper-parameters used in training and finetuning for all our downstream tasks in Section 1.2. We also provide the implementation details on the baseline methods in Section 1.3, and we show additional results, such as per-class detection and segmentation performance in Section 1.4. Finally, we show more feature visualizations in Section 1.5.

### 1.1   Model architecture

**Point-based model**   We use PointNet++ model for 3D object detection with dense point clouds. For ScanNet and SUN RGB-D, we use the same model. As shown in Table **??**, PointNet++ is consisted of four set abstraction layers and two feature up-sampling layers, as designed in [6]. Each SA layer is specified by $(n, r, [c_1, ..., c_k])$, where n represents number of output points, r represents the ball-region radius of the reception field, $c_i$ represents the feature channel size of the i-th layer in the MLP. Each feature up-sampling (FP) layer upsamples the point features by interpolating the features on input points to output points. Each FP layer is specified by $[c_1, ..., c_k]$ where $c_i$ is the output of the i-th layer in the MLP.

**Voxel-based model**   We use the popular 3D U-Net with sparse computations for efficiency improvements [2; 1; 8; 10]. The input point cloud is voxelized using a regular voxel grid (at a resolution of $0.1 \times 0.1 \times 0.1m$ for SemanticKITTI and $0.1 \times 0.1 \times 0.15m$ for Waymo Segmentation). To eliminate issues with intensity calibration and scaling, we use a binary occupancy grid. The model consists of one input convolution followed by 6 layers of U-Net blocks with the standard skip links between the encode and decode branches as proposed by [1] and used by [8]. Each block consists of a 3D convolution with 2x downsampling, followed by one sub-block (batch normalization, ReLU activation, and 3D convolution layer), and subsequently two similar sub-blocks to merge the higher layer features. The final output of the network is reprojected to yield the 64-dimensional per-point features which are used during pretraining. In the finetuning or inference mode, we further add a two layer per-point MLP as an output head. The details are listed in Table 2.

### 1.2   Finetuning details

**Object detection for ScanNet and SUN RGB-D**   We use the same configurations in VoteNet [7] for finetuning. We apply Adam optimizer [4] and use a base learning rate 0.001 with a 0.1× weight

decrease at 80, 120 and 160 epochs. The model is trained for 180 epochs in total. We use a batch size of 8 for both ScanNet and SUN RGB-D. We use the same configuration for training from scratch and finetuning, and we only load the pretrained PointNet++ backbone during fine-tuning.

**Semantic segmentation for KITTI and Waymo**   We use largely the same configurations as in JS3C [8] for their segmentation network, with small modifications in number of epochs to account for the greatly reduced training dataset sizes in our finetuning experiments. We train the model for 800, 600, 600, and 500 epochs on the 1%, 2%, 5%, and 10% splits, respectively, with a base learning rate 0.001 and a 0.7x weight decay at every tenth of the training process. We use a batch size of 32 for both datasets, and as before use the same configurations for training from scratch and finetuning experiments.

## 1.3   Baseline methods implementations

For this paper, we want to demonstrate that representation learning in local patch/volume level is better than representation learning in global level. Therefore, we consider the general setting for global-level feature learning, and we are not evaluating the benefits from multi-modality (2D-3D), multi-representation (point-voxel), and etc.

**DepthContrast [9]**   As mentioned before, we only focus on the general setting. Thus, we choose to only use the within-format setting from DepthContrast to clearly compare the benefits between local and global representation learning. Our proposed approach can be adapted to across-format learning in a similar fashion. We will address that in future work. For pretraining in both dense and sparse point clouds, we use the following data augmentations: RandomCuboidCropping, Random-LocalDrop, RandomRotation, RandomFlipping, RandomScaling. We use the same configurations as in the original DepthContrast. Since there are only 1201 training instances in ScanNet, we pre-train the PointNet++ on ScanNet for 3K epochs, with a standard SGD optimizer with momentum 0.9. We also use a cosine learning rate scheduler [5] which decreases from 0.06 to 0.00006 and train the model with a batch size of 96. On KITTI ans Waymo, we use the same pretraining settings except that we pretrain for 500 epochs each.

**BYOL [3]**   For BYOL baseline, we use a voxel-based model for pretraining, and we adopt the same data augmentation from DepthContrast. For the BYOL formulation, we changed the momentum encoder from DepthContrast to a teacher network and we add a predictor module to the student network, which is the encoder from DepthContrast. We use different initialization for the teacher and student network, and the weights of the teacher network are updated with an exponential moving average (EMA) from the student's weights. The update rule is $\theta_t \leftarrow \lambda\theta_t + (1 - \lambda)\theta_s$ with a fixed $\lambda = 0.999$. We use the following loss formulation to pretrain the model:

$$L_c^{'} = \sum_{i \in N} ||z_\theta(F_i^1) - F_i^2||^2 \tag{1}$$

$N$ is the total number of training instances and $F_i^1$, $F_i^2$ is the i-th global encoding from the student and teacher network, respectively. $z_\theta$ is a predictor for the student network. We use the same pretraining parameters as in DepthContrast.

| layer name | input layer | type | output size | layer params |
|---|---|---|---|---|
| sa1 | point cloud (xyz) | SA | (2048, 3+128) | (2048, 0.2, [64, 64, 128]) |
| sa2 | sa1 | SA | (1024, 3+256) | (1024, 0.4, [128, 128, 256]) |
| sa3 | sa2 | SA | (512, 3+256) | (512, 0.8, [128, 128, 256]) |
| sa4 | sa3 | SA | (256, 3+256) | (256, 1.2, [128, 128, 256]) |
| fp1 | sa3,sa4 | FP | (256, 3+256) | [256, 256] |
| fp2 | sa2,sa3 | FP | (256, 3+256) | [256, 256] |

Table 1: **PointNet++ Network Architecture** used in Section 1.4

## 1.4 More detailed results

**Per-category results for ScanNet**  We provide the detailed per-category object detection results for 100% annotated data setting in Table 3. We see that our pretraining improves the downstream performance on most of the categories, especially on cabinet, door, and shower curtain. With our local reasoning approach, we are able to extract more distinct features even with the planar surfaces, which boosts the performance of those categories. We observe similar behaviors in SUN RGB-D.

**Per-category results for SUN RGB-D**  Similar to the ScanNet dataset, we see notable improvements on the SUN RGB-D dataset of our pretraining scheme as compared with training from scratch and the other self-supervised learning baselines. The difference is particularly notable in the *bathtub*, *sofa*, and *bookshelf* classes, which is similar to the case in ScanNet.

**Per-category results for SemanticKITTI**  We provide the detailed per-category semantic segmentation results for the 1% annotated data setting in Table 5. We see that our pretraining improves the downstream performance after finetuning on the great majority of object classes, especially on many small or rare classes including *bicyclist*, *person*, *truck*, and *traffic-signs*, where we see up to 14% improvement over training from scratch or the baseline methods.

**Per-category results for Waymo Open Dataset Semantic Segmentation**  As with SemanticKITTI, we continue to observe significant improvements across the different semantic classes on the Waymo dataset when 1% of annotations are used. The improvements are particularly notable in the *bicyclist*, *pedestrian*, and *bicycle* categories.

## 1.5 More feature visualizations

In this section, we examine the evolution of the learned features of our approach through the learning process at different epochs. In Figure 1, we see that the learned features gradually spread out in feature space while maintaining significant separation between different classes. This demonstrates the impact of our locally contrastive scheme.

| input | layer name | output dims |
|---|---|---|
| occupancy grid | conv_in | 16 |
| conv_in | block_1 | 32 |
| block_1 | block_2 | 48 |
| block_2 | block_3 | 64 |
| block_3 | block_4 | 80 |
| block_4 | block_5 | 96 |
| block_5 | block_6 | 112 |
| block_5, block_6 | upblock_5 | 96 |
| block_4, upblock_5 | upblock_4 | 80 |
| block_3, upblock_4 | upblock_3 | 64 |
| block_2, upblock_3 | upblock_2 | 48 |
| block_1, upblock_2 | upblock_1 | 32 |
| conv_in, upblock_1 | features | 64 |
| features | logits | num_classes |

Table 2: **3D U-Net Network Architecture** used in Section 1.4

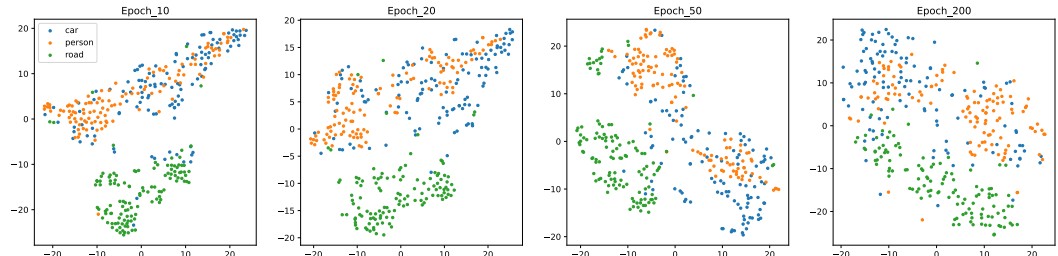

**Figure 1:** t-SNE visualization of features generated by backbone network with various weight settings.

| Pretraining | None | BYOL [3] | DepthCon [9] | Ours |
|---|---|---|---|---|
| all | 58.6 | 59.1 | 60.1 | **63.0** |
| cabinet | 36.3 | 37.4 | 35.8 | **41.2** |
| bed | 87.9 | 88.8 | 87.3 | **89.2** |
| chair | 88.7 | 88.3 | 88.8 | **90.1** |
| sofa | **89.6** | 88.4 | 88.1 | 86.8 |
| table | 58.8 | 61.4 | 63.3 | **65.4** |
| door | 47.3 | 50.1 | 50.1 | **55.1** |
| window | 38.1 | 38.7 | 41.3 | **47.3** |
| bookshelf | 44.6 | **58.9** | 57.7 | 52.9 |
| picture | 7.8 | 6.3 | 7.7 | **9.6** |
| counter | **56.1** | 50.2 | 47.1 | 49.7 |
| desk | **71.7** | 62.4 | 64.5 | 64.1 |
| curtain | 47.2 | 48.5 | 49.0 | **59.3** |
| refrigerator | 45.3 | 45.2 | 49.1 | **56.0** |
| shower curtain | 57.1 | 56.1 | 61.6 | **75.7** |
| toilet | 94.9 | 95.0 | **96.6** | 95.7 |
| sink | 54.7 | 50.9 | 55.4 | **56.0** |
| bathtub | 92.1 | 92.3 | 88.8 | **92.4** |
| garbage bin | 37.2 | 47.2 | **48.5** | 48.1 |

**Table 3: Detailed results on 100% ScanNet**

| Pretraining | None | BYOL [3] | DepthCon [9] | Ours |
|---|---|---|---|---|
| all | 57.4 | 58.0 | 59.1 | **60.1** |
| bed | 83.6 | 84.6 | 84.4 | **85.2** |
| table | 49.9 | 49.2 | 50.5 | **51.9** |
| sofa | 64.4 | 63.8 | 63.7 | **67.1** |
| chair | 74.8 | 74.9 | **75.5** | 75.3 |
| toilet | **89.8** | 88.9 | 89.7 | 88.8 |
| desk | 24.1 | 26.2 | 25.8 | **26.2** |
| dresser | 28.9 | 29.4 | **32.1** | 31.9 |
| night_stand | 59.6 | 62.4 | **63.1** | 62.1 |
| bookshelf | 30.7 | 32.1 | 34.4 | **34.5** |
| bathhub | 71.2 | 70.8 | 71.9 | **77.5** |

**Table 4: Detailed results on 100% SUN RGB-D**

**Table 5:** Detailed semantic segmentation results on 1% SemanticKITTI (mIoU)

| Pretraining | None | BYOL [3] | DepthCon [9] | Ours |
|---|---|---|---|---|
| all | 38.9 | 38.8 | 39.2 | **42.5** |
| car | 90.9 | 90.4 | 90.6 | **91.5** |
| bicycle | **3.2** | 1.8 | 1.6 | 1.1 |
| motorcycle | **5.1** | 4.4 | 4.0 | 3.7 |
| truck | 15.8 | 22.1 | 16.6 | **29.9** |
| other-vehicle | 13.1 | 15.4 | 14.3 | **23.2** |
| person | 27.7 | 24.2 | 24.2 | **34.9** |
| bicyclist | 10.7 | 9.6 | 12.6 | **17.5** |
| motorcyclist | 0.0 | 0.0 | 0.0 | 0.0 |
| road | 87.5 | 85.3 | 86.8 | **88.8** |
| parking | 22.0 | 18.3 | 22.3 | **23.2** |
| sidewalk | 66.3 | 65.0 | 65.6 | **69.5** |
| other-ground | **1.1** | 0.1 | 0.3 | 0.2 |
| building | 84.1 | 85.2 | 86.1 | **86.2** |
| fence | 33.8 | 37.5 | 39.9 | **40.5** |
| vegetation | 82.8 | 82.0 | 83.5 | **84.4** |
| trunk | 46.6 | 50.7 | 48.1 | **51.9** |
| terrain | 66.3 | 67.4 | 67.5 | **69.5** |
| pole | 43.4 | 45.2 | 49.9 | **51.5** |
| traffic-sign | 38.3 | 33.4 | 31.5 | **40.8** |

**Table 6:** Detailed semantic segmentation results on 1% Waymo Open Dataset (mIoU)

| Pretraining | None | BYOL [3] | DepthCon [9] | Ours |
|---|---|---|---|---|
| all | 42.5 | 42.3 | 42.7 | **44.8** |
| car | 88.1 | 88.8 | 89.1 | **89.8** |
| truck | 40.9 | **45.2** | 44.5 | 44.3 |
| bus | 34.2 | 37.4 | **41.1** | 37.8 |
| other vehicle | 4.5 | 5.3 | 3.5 | **6.3** |
| motorcyclist | 0.1 | **0.3** | 0.0 | 0.0 |
| bicyclist | 17.5 | 16.7 | 15.3 | **21.3** |
| pedestrian | 68.3 | 70.0 | 70.5 | **71.8** |
| sign | 49.0 | 48.7 | 49.6 | **50.0** |
| traffic light | 23.4 | **23.8** | 20.3 | 23.2 |
| pole | 54.5 | 53.6 | 53.9 | **55.1** |
| construction cone | 28.4 | 27.8 | 24.2 | **29.4** |
| bicycle | 9.9 | 7.7 | 9.4 | **18.5** |
| motorcycle | 21.2 | 19.0 | 23.5 | **24.2** |
| building | 88.9 | 88.6 | 89.1 | **89.7** |
| vegetation | 81.2 | 81.3 | 81.8 | **82.5** |
| tree trunk | 53.8 | 53.8 | 53.3 | **55.3** |
| curb | 48.4 | 47.2 | 47.9 | **50.7** |
| road | 82.2 | 83.2 | 83.1 | **85.2** |
| lane marker | 19.3 | 18.9 | 19.7 | **23.1** |
| other ground | 10.6 | 8.0 | 10.8 | **11.1** |
| walkable | 58.4 | 56.7 | 58.3 | **61.1** |
| sidewalk | 51.4 | 49.6 | 51.1 | **54.3** |