# OpenReview forum: "Self-Supervised Pretraining for Large-Scale Point Clouds"
_NeurIPS.cc/2022/Conference — NeurIPS 2022 Accept_

### Official Review · Reviewer_x6Sn · 2022-07-03

**Rating:** 6
**Confidence:** 2
**Soundness:** 2 fair
**Presentation:** 2 fair
**Contribution:** 3 good

**Summary:**

This paper proposed an unsupervised pertaining technique to improve representation learning of point clouds that can be applied to popular models for various downstream tasks.
The key idea of this paper, compared to the baseline method (DepthContrast [67]), is to learn the features from local patch level.

**Questions:**

- PointNet++ gradually downsamples the point cloud by grouping local neighbourhood together. How do you get per-point feature?
- L151, should the reference be [3] instead?


**Limitations:**

I didn't find discussions about limitations.

**Strengths And Weaknesses:**

**Strengths**
- the main idea, implemented through two different losses, i.e. local volume PointInfoNCE and global volume ClusterNCE, makes sense to me, since patch-based approach allows the feature encoding to learn from repetitive structures in large-scale point cloud scenes.
- the method is evaluated on different types of data, including outdoor dataset, which is not done in [67].
- the pretrained model can be valuable to the community.

**Weakness**
- The method seems to work well. But I have some concerns about the principles of this approach.
  - As mentioned by the authors, the information in outdoor lidar scans is very sparse, "important foreground often comprise few points". Local crops may contain zero foreground. In this case, $f^1$ does not capture any unique information that can be sufficiently associated to its corresponding global crop.
  - The pretraining takes place on local crop level and as suggested by Figure 4 the crop size matters. What happens when at test-time the scale of the input point cloud changes? In this case the receptive field of the network (e.g. the spatial span of KNN in PointNet++) also changes. How can the proposed method address such scale variation?
- The paper is quite difficult to read for anyone not up-to-date with unsupervised learning. I find Section 3.3 particularly difficult to understand. IMHO, some basic concepts of unsupervised feature clustering should be given to make the paper complete.
- Similarly, at several places the authors mentioned the advantage of local feature learning in terms of computational cost, e.g. L127-128, L141 and L146. However, from the paper itself, it's not obvious what the baseline is, why it won't scale well with the large number of occupied volumes. The authors should briefly summarise what would be the computational cost for standard practice.
- The comparison with [67] was not very clear to me. In L227-229, the author mentioned they had to reproduce [67] using a different training set, which seems to result in a slightly worse result for [67] compared to the what's originally reported (Table 2 in [67]). The reason for this isn't very clearly conveyed.

---

> ### Author Response · Authors · 2022-08-01
> **Response to Reviewer x6Sn**
>
> Thank you for your insightful review! Please find below our responses to your questions and concerns.
>
> **Q1: As mentioned by the authors, the information in outdoor lidar scans is very sparse, "important foreground often comprise few points". Local crops may contain zero foreground. In this case, does not capture any unique information that can be sufficiently associated to its corresponding global crop.**
>
> In the outdoor LiDAR scans, we note that the concept of foreground vs background is defined by annotators. Even regions that do not fall within a narrow “foreground” scope (e.g. vehicles and pedestrians) still contain meaningful information with which the model can learn useful feature descriptors. We actually observed that some global clusters are matched with some local features in almost every scene in the dataset, and they usually contain crops from roads, walls or vegetations from those clusters. Therefore, we do believe that the model is able to learn meaningful geometric features even from background points. We have edited our draft to make this point clearer to avoid confusions.
>
> **Q2: The pretraining takes place on local crop level and as suggested by Figure 4 the crop size matters. What happens when at test-time the scale of the input point cloud changes? In this case the receptive field of the network (e.g. the spatial span of KNN in PointNet++) also changes. How can the proposed method address such scale variation?**
>
> Because the method targets real world 3D point clouds, the relative consistency of physical features in the scenes means that the scales of input features to the model are usually fairly stable.
>
> **Q3: The paper is quite difficult to read for anyone not up-to-date with unsupervised learning. I find Section 3.3 particularly difficult to understand. IMHO, some basic concepts of unsupervised feature clustering should be given to make the paper complete.**
>
> We appreciate with this feedback and will add a high level overview of unsupervised feature clustering based on existing literature to the final revision of the paper to facilitate easier reading. Please check the revised paper for the initial draft.
>
> **Q4: Similarly, at several places the authors mentioned the advantage of local feature learning in terms of computational cost, e.g. L127-128, L141 and L146. However, from the paper itself, it's not obvious what the baseline is, why it won't scale well with the large number of occupied volumes. The authors should briefly summarise what would be the computational cost for standard practice. **
>
> With global feature queue size of 300k and one global feature vector from each scene, we are able to perform the InfoNCE loss with a batch size of 12 on a V100 GPU with 16GB memory size. However, with our setting, each scene contains 300 volumes on average, and each volume produces a feature vector. With the original InfoNCE loss, the computation cost is increased by 300X, and it costs too much to train such models. We will add this clarification to the final revision of the paper. Please check the revised paper for the initial draft.
>
> **Q5: The comparison with [67] was not very clear to me. In L227-229, the author mentioned they had to reproduce [67] using a different training set, which seems to result in a slightly worse result for [67] compared to the what's originally reported (Table 2 in [67]). The reason for this isn't very clearly conveyed.**
>
> In [67], their model is pre-trained with 190K single-view depth maps from ScanNet, which are extracted from RGBD video sequences. Since we mainly focus on large scale point clouds, we pre-train our model on only 1.2k point clouds on the full room scans. Therefore, we reproduce their results with our training data for fair comparison. The performance drop is expected since in [67], they stated that their pre-training scales relatively with different sized training dataset.
>
> **Q6: PointNet++ gradually downsamples the point cloud by grouping local neighbourhood together. How do you get per-point feature?**
>
> For each point in the downsampled point cloud, PointNet++ groups local neighbourhood point features, fowards the concatenated features with few MLP layers, and then conducts max-pooling. After the max-pooling, we do acquire the per-point features.
>
> **Q7: L151, should the reference be [3] instead?**
>
> Thanks for the correction. We will update the citations in the final revision. Please check the revised paper for the initial draft.
>
> **Re: Limitations:**
>
> We have added a discussion of limitations to our conclusion section.

---

> > ### Comment · Reviewer_x6Sn · 2022-08-05
> > **Thank you**
> >
> > The response clarified some of my questions. I have raised my initial score accordingly.

---

### Official Review · Reviewer_4aUz · 2022-07-10

**Rating:** 6
**Confidence:** 3
**Soundness:** 3 good
**Presentation:** 3 good
**Contribution:** 2 fair

**Summary:**

The paper proposes a self-supervised pretraining method for large-scale point clouds (SSPL). SSPL is motivated by the observation that most 3D scenes don’t consist of a single, large, foreground object but instead contain many objects of equal scale and importance. To account for this, SSPL introduces a contrastive term over local crops in the same scene and couples this to a standard global contrastive loss. SSPL implements this algorithm using standard techniques from the SSL literature. The resulting algorithm outperforms previous SSL algorithms on several 3D datasets for object detection and semantic segmentation.


**Questions:**

- How dependent is the performance of the method on the augmentation used? Augmentation is the trick by which contrastive SSL methods of this kind are given information about the structure of the target domain. So additional ablations or analysis would be helpful for interpretation of how general this method is. In particular, it would be very useful to see an error analysis as a function of the augmentation used: what specific examples in the dataset benefit from using e.g. RandomScaling vs. RandomLocalDrop?

- L273: “We see that the embeddings produced using DepthContrast and our approach are both able to separate the three classes in feature space to a much stronger degree than the BYOL baseline,...” The t-SNE plots shown in Figure 3 aren’t sufficient to support this claim. The separability exhibited in t-SNE is well-known to depend on details of the hyperparameters in unpredictable and unreliable ways (https://distill.pub/2016/misread-tsne/). I suggest reporting quantitative numbers on clustering or removing this claim. If using t-SNE plots to compare methods, it is better practice to sweep best hypers individually for each plot to avoid confirmation bias.

- L137: “pre-text” -> “pretext”

- The use of “SK” as an abbreviation for SemanticKITTI is a bit distracting and makes the paper harder to read. It makes complete sense to use this when space is severely constrained (e.g.  Table 2), but please write out SemanticKITTI in the main text if at all possible.

- Table 1: mAP@0.25 -> AP25 to match the text (L219)


**Strengths And Weaknesses:**

**Strengths**

- The paper addresses an important problem: developing SSL methods beyond the simple, highly stereotyped settings (ImageNet, single-frame ScanNet) for which standard global SSL techniques are engineered.

- The proposed method is straightforward and elegantly reuses features for both local and global contrastive losses, which in principle (and at adequate scale) should allow it to learn features that are strictly more discriminative than local- or global-only contrastive methods.

- The approach is well-motivated both in terms of the literature and in terms of the limitations of existing SSL methods for representation learning on point-clouds.

- The approach produces compellingly strong results against fair SSL baselines on several competitive benchmarks in the point cloud domain.


**Weaknesses**
- Table 1: only SSL methods are presented. These are problems that are well-addressed by existing supervised learning methods, so baseline and SotA supervised results are stronger baselines: SotA supervised on ScanNet is >70% AP25. These numbers should be reported to make it clear that while SSPL narrows the gap, it still performs significantly worse than supervised methods.

- The proposed method makes few assumptions that are specific to point clouds (other than the augmentation strategy). It would be fantastic to see this work applied to other domains, such as images (where most SSL methods are currently evaluated), audio, text, multimodal, or other domains. Is there an algorithmic reason to focus on point clouds exclusively? If not, why is the evaluation limited to point clouds, which are less competitive than other SSL domains?

- While the method is evaluated on two tasks (*detection* on ScanNet/SUN and semantic *segmentation* on SemanticKITTI and Waymo Open Dataset) it is not evaluated on multiple tasks on a single dataset. A key reason to investigate SSL at all (rather than directly optimize a task of interest, which works on this data) is that it allows direct application to multiple downstream tasks. The case for this method would be much stronger if it led to simultaneous improvements on several of the tasks supported by ScanNet, rather than just one.

---

> ### Author Response · Authors · 2022-08-01
> **Response to Reviewer 4aUz**
>
> Thank you for your insightful review! Please find below our responses to your questions and concerns.
>
> **Q1: Table 1: only SSL methods are presented. These are problems that are well-addressed by existing supervised learning methods, so baseline and SotA supervised results are stronger baselines: SotA supervised on ScanNet is >70% AP25. These numbers should be reported to make it clear that while SSPL narrows the gap, it still performs significantly worse than supervised methods.**
>
> As our focus is on model pretraining techniques through self-supervised learning, we opted to use some of the most popular and generalizable backbone architectures. We acknowledge that the current SotA exceeds the performance of our model using more involved architectures, and note that our model can be easily adapted to pretrain those models. Since H3DNet[68] shares a similar backbone with the baseline we used (VoteNet), we have also tried pretraining with H3DNet backbones. We were able to achieve a 3% boost on ScanNet detection with this setting. We will work on self-supervised pretraining for SotA models with more complex pipelines in the future.
>
> **Q2: The proposed method makes few assumptions that are specific to point clouds (other than the augmentation strategy). It would be fantastic to see this work applied to other domains, such as images (where most SSL methods are currently evaluated), audio, text, multimodal, or other domains. Is there an algorithmic reason to focus on point clouds exclusively? If not, why is the evaluation limited to point clouds, which are less competitive than other SSL domains?**
>
> In our work, our priority was to explore the setting of large scale point clouds to improve perception for robotics and self-driving vehicles. We thank the reviewer for the insightful suggestion that this technique can reasonably be extended to other domains such as large scale satellite images, and plan to explore this area in the future.
>
> **Q3: While the method is evaluated on two tasks (detection on ScanNet/SUN and semantic segmentation on SemanticKITTI and Waymo Open Dataset) it is not evaluated on multiple tasks on a single dataset. A key reason to investigate SSL at all (rather than directly optimize a task of interest, which works on this data) is that it allows direct application to multiple downstream tasks. The case for this method would be much stronger if it led to simultaneous improvements on several of the tasks supported by ScanNet, rather than just one.**
>
> We conducted a fast experiment with ScanNet semantic segmentation using the 3D UNet backbone also used for the SemanticKITTI and Waymo Open Dataset. We see +1.4%, +0.7%, and +0.4% performance gains (mIoU) for the 5%, 10%, and 20% labeled data settings, respectively via pre-training using our technique followed by finetuning compared with training from scratch. Please note that these results are obtained without **any** hyperparameter tuning, due to time constraints. We plan to add a more thorough analysis to our supplementary materials.
>
> **Q4: Regarding ablation study on augmentation used**
>
> We follow the same data augmentation schemes from an existing work, DepthContrast. DepthContrast has already established an ablation study about the effects on different data augmentation techniques. Therefore, we did not conduct these experiments for this draft. However, we do believe it may exhibit different effects on our method, and we will conduct the ablation study and append them to the supplemental materials.
>
> **Q5: Concerns about t-SNE plots**
>
> The t-SNE plots are used to visualize the feature space to get some visual intuition about the features learned. We varied the hyperparameters for each plot, and note that the qualitative comparisons for the feature distributions remain similar to that observed in the main paper. However, we agree that quantitative analysis is more rigorous. As such, we computed the feature cluster distances for each setting (randomly initialized, pre-trained with BYOL, DepthContrast, and our approach). We found that the average cluster separation is 7.3E-5, 0.02, 0.63, and 0.63 for the four settings, respectively. This backs up the claim in our paper that DepthContrast and our approach are both able to separate semantic classes to a greater degree than BYOL.
>
> **Q6: Various formatting / wording issues**
>
> Thank you for the detailed corrections, and we have addressed them in the revised paper. We will fix the listed minor issues in the camera-ready version, as well.
>
> **Re: Limitations**
>
> We have added a discussion of limitations to our conclusion section.

---

### Official Review · Reviewer_2B4N · 2022-07-11

**Rating:** 6
**Confidence:** 4
**Soundness:** 3 good
**Presentation:** 3 good
**Contribution:** 3 good

**Summary:**

The paper proposes a method for unsupervised pretraining of models for large-scale point cloud processing. The learned representations can be used for finetuning across datasets for various tasks such as point cloud classification or segmentation. The experiments show that pretraining the representation with local and global context improves performance compared to training a model from scratch.


**Questions:**

* In the experiments, all datasets contain similar scenes, so the global features encode some common context between scenes. What happens if the context is changed drastically? For example, from urban outdoor environments to more rural ones with less man-made features and more natural ones? Would this method suffer in these cases?

* Line 265: "On the other hand, we observe that the gains are reduced when additional annotated data is provided, as seen in the case with 5% and 10% annotations, as expected." Why is this expected? Please elaborate.


**Limitations:**

As far as I can tell limitations of the approach are not discussed in the paper. Consider discussing limitations in the conclusion section.

**Strengths And Weaknesses:**

Strengths:
The idea of transfer learning has been shown to be highly effective for learning in the 2D image domain. Adapting this concept to 3D data is an exciting and promising research direction. The experiments also seem to validate that unsupervised pretraining can lead to better-separated feature spaces and improve performance on downstream tasks.

* The paper is clearly written and easy to follow, well polished, and does not contain many errors. The paper also contains supplemental material with additional details for reproducibility and additional results.

* The paper motivates the proposed approach by showing a gap in the literature, focusing mainly on data containing a single prominent object. In contrast, the paper proposes to study the problem of contrastive pretraining of large-scale point clouds containing multiple objects.

* The provided ablations and visualizations of the t-SNE embeddings are very helpful in solidifying the point that the learned features are more informative and better separated in feature space than the features learned from scratch or by the baselines.

Weaknesses:
* As I am not intrinsically familiar with the literature on contrastive learning on point clouds, it was unclear what the intuition was behind global features. Here are some pointers towards making this clear:
1. In Figure 2. what is the dimension of the global feature queue?
2. In Section 3.3: What is the intuition behind global features? What do they represent? How is the global feature queue constructed and updated? Are random f2 features from the momentum encoder added to this queue or do some criteria choose them?

* One point the paper should address is how the proposed method compares against methods that use diverse, labeled datasets for pretraining. I think it would be interesting to see how the proposed method performs against a method that, for example, uses SUN and ScanNet training sets for pretraining. I am especially interested if the proposed method's resulting feature space is still more informative (I think it might be).

* Code and data are not released with this paper. Is there a reason why the paper does not contain this or will this be released upon acceptance?

Minor issues:
* Algorithm 1: "Computer local contrastive" -> Compute local contrastive
* Line 275: "the the" -> the
* Line 278: "This is hypothesis" -> This hypothesis

---

> ### Author Response · Authors · 2022-08-01
> **Response to Reviewer 2B4N**
>
> Thank you for your insightful review! Please find below our responses to your questions and concerns.
>
> **Q1. In Figure 2. what is the dimension of the global feature queue?**
>
> We set the size of global feature queue to be 300K.
>
> **Q2. In Section 3.3: What is the intuition behind global features? What do they represent? How is the global feature queue constructed and updated? Are random f2 features from the momentum encoder added to this queue or do some criteria choose them?**
>
> The global features are volume features extracted across all data instances in the dataset. They contain local geometry features from different scenes. The global feature queue is updated with a first-in-first-out scheme. When updating the queue, we append f2 features from the momentum encoder directly to the queue. The clustering and matching is done during the loss function optimization.
>
> **Q3: One point the paper should address is how the proposed method compares against methods that use diverse, labeled datasets for pretraining. I think it would be interesting to see how the proposed method performs against a method that, for example, uses SUN and ScanNet training sets for pretraining. I am especially interested if the proposed method's resulting feature space is still more informative (I think it might be).**
>
> In our paper, we focus on the fully self-supervised learning setting, where no supervised data is available for pre-training. This is common with most other self-supervised learning papers. However, comparing with using supervised data for pre-training in addition to or instead of our approach is an interesting future research direction.
>
> **Q4: Code and data are not released with this paper. Is there a reason why the paper does not contain this or will this be released upon acceptance?**
>
> The data used in our work is publicly available. We plan to release code upon acceptance. Moreover, we believe that we have made details required for reproduction available in our paper and supplementary materials.
>
> **Q5: Algorithm 1: "Computer local contrastive" → Compute local contrastive, Line 275: "the the" → the, Line 278: "This is hypothesis" → This hypothesis (Fix the paper in revisions)**
>
> Thanks for the correction, and we have addressed them in the revised paper. We will fix the listed minor issues in the camera-ready version, as well.
>
> **Q6: In the experiments, all datasets contain similar scenes, so the global features encode some common context between scenes. What happens if the context is changed drastically? For example, from urban outdoor environments to more rural ones with less man-made features and more natural ones? Would this method suffer in these cases?**
>
> If the context of the dataset is more diverse, e.g. from urban outdoor to rural environments, we can just increase the global queue size and maybe increase the number of clusters globally. With more diverse data, the features tend to form more cluster, resulting in a need to increase the number of clusters and the global queue size.
>
> **Q7: Line 265: "On the other hand, we observe that the gains are reduced when additional annotated data is provided, as seen in the case with 5% and 10% annotations, as expected." Why is this expected? Please elaborate.**
>
> We have seen in existing self-supervised learning papers (such as SimCLR) that the performance gap between training a neural network from scratch vs initializing from a the result of self-supervised learning tends to shrink as the amount of annotated data increases. Intuitively, with a very large amount of directly applicable labeled data to directly optimize the model’s performance on downstream tasks, the amount of additional information a model can learn from auxiliary tasks (such as those in self-supervision) decreases.
>
> **Re: Limitations**
>
> We have added a discussion of limitations to our conclusion section.

---

### Meta-Review · Area_Chair_FAjG · 2022-08-27

**Recommendation:** Accept
**Confidence:** Less certain

**Metareview:**

All three reviewers recommend acceptance after rebuttal, although they were not very confident. I also read the paper and agree it is well written, has good results and addresses a gap in the large-scale point cloud learning literature. However I also see that the paper selectively builds upon ideas from other domains, e.g. vision [19,3] while ignoring closest related work that also does self-supervised learning in images from large scenes, using also local or local+global contrastive learning. Papers that are highly related include:

- Contrastive learning of global and local features for medical image segmentation with limited annotations. Chaitanya et al.
- Efficient Visual Pretraining with Contrastive Detection, Henaff et al.
- Point-Level Region Contrast for Object Detection Pre-Training. Bai et al.
- SegContrast: 3D Point Cloud Feature Representation Learning Through Self-Supervised Segment Discrimination. Nunes et al.

I strongly suggest explaining the differences to these papers. In an ideal world the first one should be a baseline to really show if there's something about the fit of the proposed method and point cloud data.

**Award:**

No

---

### Decision · Program_Chairs · 2022-09-14

Accept